# Is There an Interconnection between Epithelial–Mesenchymal Transition (EMT) and Telomere Shortening in Aging?

**DOI:** 10.3390/ijms22083888

**Published:** 2021-04-09

**Authors:** Siti A. M. Imran, Muhammad Dain Yazid, Ruszymah Bt Hj Idrus, Manira Maarof, Abid Nordin, Rabiatul Adawiyah Razali, Yogeswaran Lokanathan

**Affiliations:** 1Centre for Tissue Engineering and Regenerative Medicine, Faculty of Medicine, Universiti Kebangsaan Malaysia, Jalan Yaacob Latiff, Bandar Tun Razak, Cheras, Kuala Lumpur 56000, Malaysia; siti.imran@ukm.edu.my (S.A.M.I.); dain@ukm.edu.my (M.D.Y.); ruszyidrus@gmail.com (R.B.H.I.); manira@ppukm.ukm.edu.my (M.M.); m.abid.nordin@gmail.com (A.N.); rabiatularzl@ukm.edu.my (R.A.R.); 2Department of Physiology, Faculty of Medicine, Universiti Kebangsaan Malaysia, Jalan Yaacob Latiff, Bandar Tun Razak, Cheras, Kuala Lumpur 56000, Malaysia

**Keywords:** EMT, shelterin, senescent, age-related disease, telomere protection

## Abstract

Epithelial–Mesenchymal Transition (EMT) was first discovered during the transition of cells from the primitive streak during embryogenesis in chicks. It was later discovered that EMT holds greater potential in areas other than the early development of cells and tissues since it also plays a vital role in wound healing and cancer development. EMT can be classified into three types based on physiological functions. EMT type 3, which involves neoplastic development and metastasis, has been the most thoroughly explored. As EMT is often found in cancer stem cells, most research has focused on its association with other factors involving cancer progression, including telomeres. However, as telomeres are also mainly involved in aging, any possible interaction between the two would be worth noting, especially as telomere dysfunction also contributes to cancer and other age-related diseases. Ascertaining the balance between degeneration and cancer development is crucial in cell biology, in which telomeres function as a key regulator between the two extremes. The essential roles that EMT and telomere protection have in aging reveal a potential mutual interaction that has not yet been explored, and which could be used in disease therapy. In this review, the known functions of EMT and telomeres in aging are discussed and their potential interaction in age-related diseases is highlighted.

## 1. Introduction

Aging is associated with a range of related diseases. As humans age, they are more prone to developing multiple diseases. This has led researchers to further investigate the relationship between aging and age-related diseases. Does disease occur because the body is regressing due to the decay in the organ system and the decline of normal physiological functions?

Disease and aging are closely related to epithelial–mesenchymal transition (EMT) and telomeres. A link between EMT and aging has been established, in that EMT has been found to play a crucial role in the age-related development of fibrosis in the heart and lungs [1]. Genes associated with EMT, such as transforming growth factor-β (TGF-β), have also been found to increase in the brain of patients with Alzheimer’s Disease (AD), which causes chronic neuroinflammation [2]. Aging also increases TGF-β expression, which induces EMT [3,4]. The upregulation of TGF-β could contribute to cell degeneration, tissue fibrosis, inflammation, decreased regeneration capacity, and metabolic malfunction, which will, in turn, lead to the development of various types of diseases [5,6] including cancer [7,8]. Telomeres, on the other hand, have been long associated with aging since they were first discovered [9,10]. The relationship between telomere length and aging as well as age-related diseases has been broadly explored, whereby many previous studies have reported that short telomere length is the cause of various diseases, such as dyskeratosis congenita [11,12,13], pulmonary fibrosis [14,15], and even cancer [16,17,18,19].

The relationship between telomeres and EMT has not yet been established; in fact, it has not been examined by any research studies so far. However, the interconnecting function that links both aging and cancer development may indicate a possible correlation between the two, which might enable further understanding of the mechanism to develop future therapies. As shown in Figure 1, through aging, telomere length decreases while EMT-associated proteins levels increase, which shows a possible correlation between the two. Understanding this relationship might enable the discovery of potential therapies for various age-related diseases involving EMT and telomere length maintenance. This review aims to discuss the known functions of EMT and telomeres in relation to aging as well as the possible interaction between the two in terms of age-related diseases.

## 2. EMT

A vital aspect in embryogenesis is the cell conversion from epithelial to mesenchymal cells, especially in tissue and organ growth. The biological sequence of EMT involves the differentiation of epithelial cells into mesenchymal phenotype cells with a greater capacity for migration and an invasive character. EMT-induced cells resist apoptosis in an enhanced manner and significantly increase extracellular matrix (ECM) component production. In addition to its role in embryogenesis, the EMT process may be activated to heal wounds and regenerate tissue. The key aspects of epithelial cell characteristics are the adhesion between the cells and the apical–basal polarity. These are created due to the way tight junctions, adherens junctions, gap junctions, and desmosomes are arranged [20]. Cells of this type are located on a basement membrane, forming at least one layer that functions as a barrier to the outlines of the tissues and organs.

EMT is induced by numerous transcription factors and signaling pathways, depending on both physiological and pathological circumstances. As the most potent EMT activator, TGF-β leads signaling pathways to become activated. This process culminates in the expression of genes with the function of encoding EMT transcription factors (EMT-TFs) [4,21,22]. In EMT, three major groups of EMT-TFs play vital roles, including SNAI (Snail and Slug), ZEB (ZEB1 and ZEB2) and TWIST (TWIST1 and TWIST2) [23,24,25,26], which repress the expression of E-cadherin, thus causing the disassembly of cell–cell junctions and inducing EMT [27,28]. As the main component, TGF-β is also involved in many signaling pathways contributing to the development of EMT, such as the TGF-β signaling pathway [29], the WNT signaling pathway [4,30], and the Smad Signaling pathway [31,32], which regulates the EMT process. Furthermore, growth factors such as insulin-like growth factor (IGF), fibroblast growth factor (FGF) and epidermal growth factor (EGF) can also trigger EMT via the previously-mentioned transcription factors [33,34].

Although the specific roles and triggers of EMT remain unclear, it can be classified into three different types according to biological function. EMT is associated with implantation, embryo formation, and organ development, and is organized to generate diverse cell types that share common mesenchymal phenotypes. This class of EMT, which has the proposed term ‘type 1,′ neither causes fibrosis nor induces an invasive phenotype resulting in systemic spread via circulation. This type of EMT occurs during embryonic gastrulation that gives rise to the mesoderm and endoderm. The primitive epithelium during this stage could generate the primary mesenchyme (mesenchymal cells) through EMT, and could also be reverted to secondary epithelial via mesenchymal–epithelial transition (MET) [35].

The second type of EMT is associated with wound healing, tissue regeneration, and organ fibrosis. Type 2 EMT involves tissue reconstruction, required due to trauma or inflammatory injury, which induces a repair cascade that normally generates fibroblasts and other related cells. Unlike type 1 EMT, type 2 EMT is not naturally occurring and is linked to inflammation. As inflammation is attenuated, EMT type 2 reduces, as observed in wound healing and tissue regeneration. Type 2 EMT can provide a continual response to ongoing inflammation in organ fibrosis. However, extended expression of EMT factors may cause organ damage. Persistent inflammation causes prolonged wound healing, known as tissue fibrosis.

Type 3 EMT occurs in neoplastic cells that undergo a genetic and epigenetic change; this refers specifically to genes in which clonal outgrowth is favored and localized tumors are developed. Such alterations, which particularly affect oncogenes and tumor suppressor genes, when acting in cooperation with the regulatory circuitry of EMT, will produce vastly dissimilar results from those witnessed with the two other forms of EMT. If the cells undergo a type 3 EMT, carcinoma cells metastasize and invasion might occur, thus increasing the progression of cancer. Significantly, the extent to which cancer cells might traverse EMT differs. Certain cells retain numerous epithelial attributes as they acquire certain mesenchymal aspects, while different cells lose all remnants of their epithelial origins and become completely mesenchymal. The exact signals that induce type 3 EMT in carcinoma cells remain unexplained. Numerous primary carcinomas may be linked to various signals originating from tumor stroma.

These EMT classifications could enable further comprehension of EMT function in biological processes and assist in investigations related to the mechanisms involved in distinguishing each type of EMT. Most studies have only focused on type 1 and type 3 EMTs, as the function in these cases is crucial in embryogenesis as well as cancer development. However, an exploration of type 2 EMT potentially offers a greater in-depth understanding of its role in aging and age-related diseases, since this type has been associated with aging, where EMT has the potential to regenerate and replace damaged or dead cells.

## 3. EMT and Aging

Aging can be distinguished by the decline in cell, tissue, and organ function, which is connected to a greater risk of developing age-related diseases and is also influenced by environmental and lifestyle factors [36,37]. Some research has shown that aging impacts the regenerative capabilities of tissues [38]. However, work on animal models with strong regenerative capabilities, such as zebrafish and newts, suggests that aging does not affect regeneration [39,40]. Repair and regeneration of tissues are vital processes in the maintenance of an organism’s integrity and ability to survive. When aging occurs, the reliability of these mechanisms reduces, resulting in a lower capacity for repair and an ongoing reduction in the structural and functional capacity of the tissue [41,42]. A decline in the normal function of cells is considered concomitant with aging progression and the development of age-related diseases. It has been shown that advanced age is connected to pathological fibrosis, and is a fibrotic disorder risk factor [43]. Pathological fibrosis is mediated by fibroblasts that are responsible for the deposition of ECM components. The accumulation of fibroblast cells leads to excess production of fibrotic tissues, thus compromising the normal physiological function of the vital organs [44]. It has been proven that certain fibroblasts derive from epithelial cells that have undergone EMT, indicating that this procedure plays a key role in tissue fibrosis [45]. E-cadherin is less expressed in mesenchymal fibroblasts, whereas alpha-smooth muscle actin (α-SMA) is highly expressed in mesenchymal fibroblasts of aged rats [46]. Additionally, cardiac fibrosis could be caused by an occurrence of EMT in the heart, particularly in a patient suffering from advanced cardiac failure [47]. Furthermore, fibrosis forms one feature of cardiovascular pathology in syndromes where aging is accelerated, as in for example Hutchinson–Gilford progeria syndrome (HGPS). This can be compared to the cardiovascular pathologies noted among geriatric patients [48]. In addition, older cells have been discovered with altered differential TGF-β, which bears a close resemblance to the profiles of patients suffering from HGPS [49].

In Idiopathic Pulmonary Fibrosis (IPF) patients, the nucleus has been shown to contain β-catenin [50]. This indicates that the Wnt/β-catenin pathway is involved in EMT-related fibrosis. Moreover, IPF, a characteristic of which is lost respiratory functionality caused by ECM being excessively deposited, could induce EMT, since an abnormal Wnt/β-catenin signaling pathway is exhibited in such cases [51].

Diminished function in the blood–brain barrier (BBB) properties is a major occurrence in a number of conditions, such as multiple sclerosis (MS) [52] or in normal human aging [53]. According to Troletti et al. (2016), EMT may play a role during blood–brain barrier dysfunction in neurological disorders, as EMT interacts with factors that give rise to MS pathogenesis. The collapse in the BBB has been reported to be involved in several neurodegenerative diseases such as Alzheimer’s disease [54]. EMT may also play a potential role in Alzheimer’s disease, where certain areas of the affected brain were found to have a high expression of genes associated with EMT [2].

Aging is also associated with the development of senescent cells. Senescent cells exhibit metabolic and functional changes, including cell cycle arrest. They also acquire a senescent-associated secretory phenotype (SASP) characterized by increased secretion of the pro-inflammatory phenotype secreting cytokines, growth factors, metalloproteinases, and reactive oxygen species [55]. The increase in SASP in senescent cells creates an environment that is harmful to other cells, which may contribute to age-related diseases such as arteriosclerosis [56] and cancer [57]. The induction of EMT might be influenced by senescent cells. Secretory phenotypes linked to these cells that derive from senescent fibroblasts have the capacity to initiate EMT induction in the surrounding epithelial cells [58]. The latter displayed low levels of cell membrane-associated β-catenin, E-cadherin, and cytokeratin, while the vimentin protein level rose. These represent key indicators of EMT following treatment with a conditioned medium from senescent cells. It is interesting to note that stimulation by the senescent cells-conditioned medium is reduced by inhibiting inflammatory cytokines, namely Interleukin-6 (IL-6) and Interleukin-8 (IL-8). Meanwhile, addition of IL-6 and IL-8 to the pre-senescent cell-conditioned medium appeared to result in the promotion of cancer cell invasion [59]. Inflammatory cytokines IL-6 and IL-8 as well as chemokines CXCL-1 and human growth regulated α protein (human GROα) are among the highly secreted human SASP factors [59,60]. Consequently, it seems that SASP and inflammation promote EMT. Larger amounts of hepatocyte growth factor (HGF) were identified in the senescent fibroblast-conditioned medium compared to pre-senescent prostate fibroblasts. An association exists between HGF and cell–cell junction disintegration, epithelial cell morphogenesis disruption, as well as migration and invasion stimulation; hence, in the neighboring epithelia, EMT is promoted [61,62]. Furthermore, a higher level of HGF has been identified within skin fibroblasts among aged individuals, which responds to the increased level of insulin-like growth factors (IGFs) [63]. A further alteration linked to age is the level of various growth factors, such as TGF-β, epithelial growth factor (EGF) and IGFs, which may increase EMT and fibrosis progression [64]. However, the regulatory function of EMT transcription factors, along with several senescent key players in different conditions, is still unclear.

Cancer and aging are interconnected since through the latter, the risk factor for cancer development also increases. An important tumor suppressor, p53, has been reported to decline with age [65]. However, SASP may also be targeted by p53, as it has been reported that the loss of p53 caused higher levels of SASP components, such as IL-6 and IL-8, to be secreted in the cells [60,66]. These have the ability to induce EMT, suggesting an important association of p53 with senescence [67]. The function of p53 was shown to be significantly more important in preventing cancer in older organisms; a study found a higher incidence of tumors in mice at 12 months compared to mice at 3 months after p53 deletion [68]. This is because SASP prevention by p53 becomes less effective as the levels of p53 declines with age [69]. This would, in turn, cause an accumulation of SASP components that may induce EMT, thus leading to development of tissue fibrosis or cancer.

## 4. Telomeres and Aging

Telomeres, consisting of TTAGGG repeats at the end part of the chromosome. The telomere-binding proteins, i.e., shelterin, form structures such as the T-loop and D-loop that function to protect the genomic integrity of the chromosome [70]. Shelterin, a protein complex that protects telomeres, consists of six subunits, namely telomere repeat factors 1 and 2 (TRF1, TRF2), protection of the telomere 1 (POT1), repressor/activator protein 1 (RAP1), TRF1- and TRF2-interacting nuclear protein 2 (TIN2) as well as the *ACD* gene (TPP1). Shelterin helps to protect the telomeres from being recognized as a DNA break [71].

The mitotic cells in the body replicate multiple times to regenerate dead and injured cells, some more so than others. For example, epithelial cells in the intestine and stomach replicate more frequently as these cells are shed often due to the digestive process. In contrast to embryonic stem cells, which have the ability to proliferate indefinitely in culture, terminally differentiated cells have a limited number of replication, which is known as the Hayflick limit. Such limitations are usually due to critically short telomeres in somatic cells without telomerase activity. As cells replicate, telomere length shortens by approximately 30–200 bp every cell cycle until it reaches a length that causes cells to become senescent [72]. However, telomere length is maintained by shelterin as well as telomerase [70,71]. The ability to maintain healthy replication and proliferation is the synergistic function of telomerase and shelterin in maintaining the telomere length. When this ability is disrupted, the telomere length shortens; thus, it may cause senescence when the telomere length becomes critically short. A healthy telomere length could also be disrupted due to genotoxic stresses such as chemotherapy and radiation, also known as therapy-induced senescence (TIS) [73]. Cells that have higher proliferation, such as hematopoietic progenitor cells, or keratinocytes, have higher telomerase activity, which helps in maintaining the telomere length. However, this process of telomere maintenance occurs only in cells with active telomerase while the majority of human somatic cells do not have telomerase activity [74]. In adult human stem cells, low levels of telomerase were found and were increased in cells that undergo rapid expansion [75]. Low levels of telomerase in highly proliferative cells cause the telomere length to shorten and therefore renders the cells unable to replicate, thus hindering the cell replenishment of damaged and senescent cells [76]. The amount of telomerase in human cells can be upregulated in adult stem cells, but it is not expressed in most human somatic cells except for lymphocytes and endothelial cells [77]. When the telomeres become critically short due to the lack of telomerase, the cells would either become senescent or undergo apoptosis.

However, there is a distinguishable difference between the role of telomere dysfunction in dividing cells and the stable DNA damage that occurs in telomeres of any length in non-dividing cells [78,79]. In dividing cells, telomerase is capable of restoring telomere length [80] and slowly resolving DNA damage [81,82] but there is no telomerase activity in non-dividing cells (at least in humans), whereas telomere dysfunction in these cells is thought to be permanent or at least long-lasting. Additionally, basal keratinocytes may have constitutive alternative lengthening of telomeres (ALT) activity [83] as well as telomerase activity [84] and only shorten their telomeres slightly with chronological age.

Short telomeres are also associated with the development of certain cancer types, such as breast cancer [85]. Some cells with terminally short telomeres are prone to having end-to-end chromosomal fusions, which will cause DNA instability that could lead to cancer. This may seem contraindicated, as cancer is highly proliferative and long telomeres have also been associated with a higher risk in the development of some cancers [86]. Cancer cells with short telomeres were able to proliferate due to the high levels of telomerase activity present [87]. Another pathway of cancer development is through ALT [88], which is mostly activated in cells lacking in telomerase, such as sarcomas [89]. A study has shown that ALT is associated with an upregulation of RNA TERRA [90]. A whole genome sequencing (WGS) analysis showed that there was no association of ALT with mutations of ATRX [91] even though it has been reported in some cancer types, such as human glioma [92] and prostate cancer [93]. However, the maintenance of ALT and how it is activated or regulated are poorly understood, either in cancerous or non-cancerous cells.

Previous reports suggest that the inactivation of telomerase activity accelerates aging in budding yeast [94]. Since inactivation of telomerase causes short telomere length, a lack of telomerase has been associated with aging [95]. This could explain the increasing prevalence of cancer development with age. Other than mutation, aging has always been another major factor in cancer development. Aging due to telomere shortening has been regarded as reversible through the manipulation of telomerase expression [95]. However, telomerase dysfunction, along with other subsequent DNA damage response activation, may lead to aging and cancer [17].

## 5. Is There Any Interconnection between EMT and Telomeres?

So far, from previously reported studies, no known factors play a role in both EMT and telomere length maintenance. The induction of SNAIL1 transcription factor, a key regulator of EMT in human prostate cancer cell lines, resulted in the inhibition of senescence [96]. SNAIL1 has also been reported to play an essential role in tumor progression by controlling the expansion and activity of tumor-initiating cells in pre-neoplastic glands and established tumors [97]. Collectively, these data suggest that the initiation of EMT via SNAIL1 can avoid senescence in cancerous cells. However, a different scenario occurs in non-cancerous cells. One study reported that SNAIL1 controls telomere transcription and integrity. SNAIL1-deficient mouse mesenchymal stem cells (MSCs) were found to contain higher levels of telomerase activity. SNAIL1 expression was also found to downregulate the expression of the telomerase gene (TERT) as well as the long non-coding RNA called telomeric repeat-containing RNA (TERRA) at different subtelomeric regions of the chromosome in mice (chr2q, chr11q, and chr18q) [98]. TERRA and TERT are transiently downregulated in TGFβ-induced EMT [98]. These data indicate that SNAIL1 plays a vital role in maintaining telomere length.

Based on STRING databases, both SNAIL1 and p53 were found to interact with TRF1 in the telomere. However, it remains unknown how those protein interact or are involved in the crosstalk of EMT and telomere maintenance. A group of researchers from The Wistar Institute identified that p53 was bound to the sub-telomere [99]. These are segments of DNA, situated between the telomeres and chromatin, which may interact with any sub-protein of shelterin. As depicted in Figure 2, a schematic diagram has been provided showing the interconnectivity of EMT, specifically SNAIL1 with a shelterin subprotein, namely TRF1. It is hoped that this would provide some indication of an interaction network prediction that could suggest new directions for future experimental research.

As TRF2 has been reported to interact with various factors and acts as a protein hub in regulating and protecting the telomere length [22], it may also interact with one or more factors regulating EMT. Therefore, it is necessary to explore the relationship of telomere protection factors, i.e., shelterin, by looking at their expression and their protein levels in EMT-induced cells and investigating the relationship it may play with EMT factors such as TGF-β. Both TRF2 and TGF-β play a vital role in cancer development, whereby TRF2 directly interacts with a DNA damage response pathway, such as ataxia–telangiectasia mutated (ATM) to detect DNA break and initiate a repair cascade [100]. DNA instability and mutation are the hallmarks of cancer development, whereby telomere protection is essential in shielding the telomere ends from being recognized as a DNA break, which in turn will cause DNA instability and mutation [100]. Meanwhile, TGF-β is a major regulatory factor in hormonal and immune responses, cell growth, cell death, cell immortalization, bone formation, tissue remodeling, and repair. In normal and healthy tissue, TGF-β induces tumor-suppressive effects, thus protecting the body from tumor development. However, mutation and disruption in the TGF-β signaling pathway, including its receptors, will cause tumor progression [101,102]. The relationship between TGF-β and the DNA damage response pathway has also been established during telomere attrition due to TIS [103].

Although there is no direct interaction between TRF2 and TGF-β, they may interact indirectly through other factors. For example, the TGF-β receptor has been reported to be temporally and spatially activated, and it was reported to be involved in the recruitment of stem/progenitor cells participation in the tissue regeneration/remodeling process [104]. TRF2 is also important in stem cell development, particularly in neural development, where downregulating the levels of TRF2 hinders the ability of human embryonic stem cell to differentiate into neural lineages [105]. Manipulating the levels of TRF2 in cells has also been reported to promote tumorigenicity [106,107,108,109], which is similar to the function of TGF-β in EMT, where the addition of TGF-β in epithelial cell culture will induce EMT and metastasis in cancer [4]. As TGF-β, an essential factor in EMT development, and telomere protection factors, namely TRF2, both play an important role in age-related diseases such as cancer, a direct or indirect interaction may exist between the two. DNA damage signaling at the telomeres has been reported to regulate the progression and metastasis of cancer stem cells [110]. The schematic diagram, as shown in Figure 2, suggests that Akt, a downstream factor in the TGF-β signaling pathway is directly inhibited by TRF1 [111]. The former is also involved in the pathway for DNA damage response [112]. Telomere dysregulation has also been reported to cause a DNA damage response, thus inducing the development of pulmonary fibrosis and increasing the expression of EMT-associated marker [113]. These similarities suggest a possible overlap between TRF2 and TGF-β in the regulation and maintenance of normal physiological functions as aging is a complex process involving multiple factors that could cause DNA instability, which in turn would promote the development of age-related diseases or cancer [114].

## 6. Future Directions

Although the relationship between telomere protection and EMT has not yet been fully explored, the overlap between factors involved in both mechanisms, such as SNAIL, TRF1, TRF2, and TGF-β, reveals a possible interaction between the two important mechanisms in terms of aging and age-related diseases. This could add more information on the already-existing connection between senescence and EMT. As telomere protection and EMT can be considered the gatekeepers for age-related diseases, discovering the mechanism that interconnects the two might lead to the possibility of developing potential forms of therapy to overcome such diseases.

## Figures and Tables

**Figure 1 ijms-22-03888-f001:**
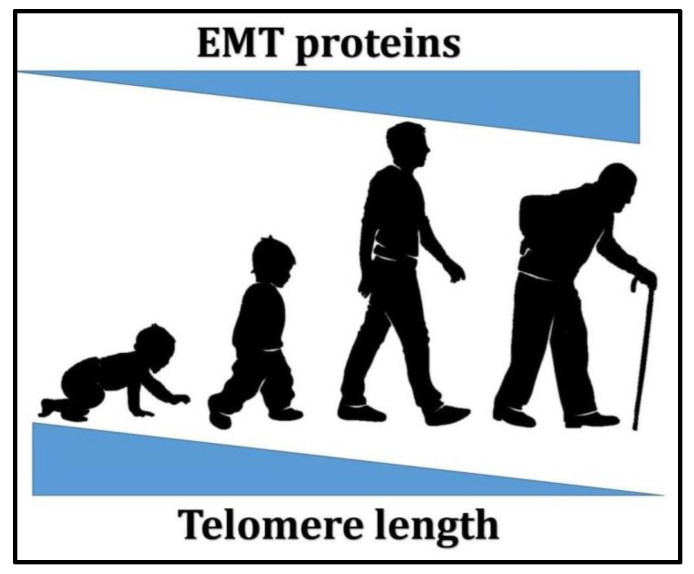
A schematic representation of the relationship between aging, telomere length, and epithelial–mesenchymal transition (EMT)-associated proteins levels.

**Figure 2 ijms-22-03888-f002:**
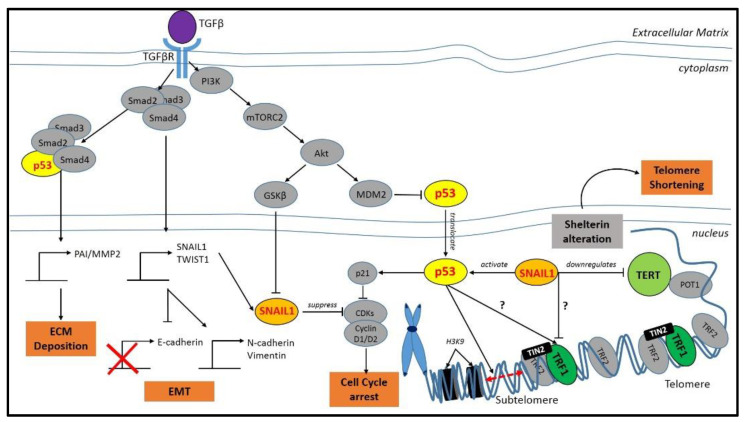
A schematic diagram of the interconnection between EMT and TRF1/2. The cross sign indicates downregulation of the protein and the question mark indicates that the pathway or regulation is still unknown. Apart from EMT response, activation of the transforming growth factor-β (TGF-β) signaling pathway coordinates various critical events such as extracellular matrix (ECM) deposition and cell cycle arrest. It is predicted that EMT is also involved in telomere shortening via SNAIL1–TERT–TRF1 interaction.

## Data Availability

No new data were created or analyzed in this study. Data sharing is not applicable to this article.

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
