# Peer review of "Is There an Interconnection between Epithelial–Mesenchymal Transition (EMT) and Telomere Shortening in Aging?"

_ijms, 2021, doi:10.3390/ijms22083888_

Round 1

Reviewer 1 Report

This is a potentially interesting and rather well-written review. However, without having any real scientific evidence on any potential connection between EMT and telomeres in the context of ageing, the topic seems a bit far-fetched and constructed since no studies exist in this area and all the authors present or hypothesises and some connection between TRF2 and TGFbeta which again, seems pretty constructed and far fetched since you can make the claim that in biology almost everything is connected between each other.

Some specific comments (including language and grammar):

  1. There is not only no literature on a potential association of EMT and telomeres in ageing, also there is hardly any for cancer as well. And the one there is Telomere DNA damage signaling regulates cancer stem cell evolution, epithelial mesenchymal transition, and metastasis. Lagunas AM, Wu J, Crowe DL. Oncotarget. 2017 the authors have not even mentioned. So please change the abstract accordingly-you cannot say that there is "a lot of research", this is simply not true.
  2. I suggest that also in the title that should be highlighted to make clear that there is no real interconnection between EMT and telomeres in ageing, just a potential or hypothetic without any real scientific evidence, otherwise this is misleading the reader.
  3. Please use "telomeres" in plural for most cases since there are 92 per normal cell.
  4. In the abstract line 17: What is "stem cell cancer"-this is a wrong term and has to be corrected. Do you mean "Cancer stem cells"?
  5. Introduction, line 29: The sentences "Aging is THE hallmark FOR diseases" is wrong. Perhaps: "aging is a hallmark of age-related diseases"? There are plenty of diseases not connected to age. Also "These progression" is both wrong in grammar ("THIS progression") and content: You have not described ANY progression yet-just mentioned "disease". Please correct.
  6. line 52: grammar should be correct "protein expression", but this term is wrong since ONLY GENES are expressed-not RNA or proteins. Please correct.
  7. line 78 should be: "...contribute to the disassembly"...
  8. line 79: it should be: "TGFbeta IS also involveD"...
  9. line 126/7: about Aging impacting tissue regeneration can also be seen the other way-that a decline in adult stem cell functionality is the main reason for a decrease in tissue regeneration and therefore a cause of ageing: to my knowledge there is no real consense about the "hen and egg" question. Thus, be more careful with such oversimplified statements.
  10. line 134: "fibroblastS that ARE responsible..."-plural is required here.
  11. lines 158/9: another wrong statement: It is NOT true that breakdown of BBB CAUSES AD, there are other brain-internal processes like the accumulation of beta amyloid and tau hyperphosphorylation. BBB might somehow be involved but is not causal. Please amend and correct the statement.
  12. Likewise, for the statement about EMT genes in AD (line 161) you need to provide a proper reference.
  13. When talking about senescence in the paragraph starting in line 162, please bear in mind that senescence does not only occur in dividing/mitotic, but also in postmitotic cell, just without a cell cycle arrest, but SASP and other factors such as damaged (though not shortened) telomeres also occur there.
  14. lines 177/8: again, grammar and content mistakes. "In the prostate", but this is an organ and you talk about conditioned serum and cell culture where a "prostate" as a complex human organ is certainly not found. Please correct the statement. perhaps you mean "prostate (cancer?) cells? This is a serious difference!
  15. I detected some plagiarism: I found at least a 100% identical sentence (lines 181-3) with a recent review on EMT and aging (Santos et al., 2019). This raises the suspicion that there might be more and this is worrying! In addition, please explain "IGF".
  16. line 186: the grammar in "function...with" is not correct, perhaps "function... FOR".
  17. 189: some sentences are a bit clumsy in language. You could here just state that "p53 is declined..." "be declined" is in any case wrong and the active form preferable.
  18. line 191, please correct SAPS to SASP
  19. line 197 the formulation "p53 prevention of SASP" is not correct. Do you mean "p53 prevents SASP"? Some more explanation about the mechanism should be given since it seems like one of those many wild speculations without any evidence. Just the fact that many functions and factors decline with aging does not mean that those are causally linked or interacting.
  20. lines 201-207 are INSTRUCTIONS for the manuscript and should NOT be included under "TelomereS and Aging"-telomeres require plural most of the time unless you talk about a molecular mechanism referring just to a single telomere.
  21. line 207 the sentence "The telomere is the repeats" is wrong in grammar and content. Telomeres consist of DNA (repeats) AND telomere-binding proteins (shelterin, as correctly described) and form a higher-order structure including a T- and D-loop. ALL this ARE telomeres. Please correct.
  22. line 213: here you compare dividing (intestinal etc) and postmitotic (cardiac) cells and say the former divide more than the latter: THIS is the difference between mitotic and postmitotic cells!
  23. same in line 215/6: You have to make sure to state what type of stem cells you refer to since ONLY embryonic stem cells with high telomerase activity (TA) IN CULTURE (not naturally where they differentiate during embryogensis) are able to proliferate INDEFINETELY, not "infinitely" as you wrote (also wrong in lines 227 and 245. In contrast. Adult stem cells have limited amounts of TA and proliferation which both decline with age as does their functionality. Reference 72 refers to adult stem cells (HSCs)...
  24. In line 228: it should be "helps"-its still does, not only in the past.
  25. lines 244/5 Your statement about cancer cells requiring long telomeres is WRONG! Most cancer cells have short telomeres, but high levels of TA maintaining their proper capping and therefore also the indefinite proliferation capability of cancer cell. Please correct.
  26. line 247: please replace "senescent" with "senescence" and the statement that ALT is activated in cells with short telomeres is to my knowledge not really true (please provide a proper reference for it if you think it is correct). Cells with ALT are characterised with extremely heterogenous telomere lengths (TL). Also, ALT maintains telomeres in fertilised oocytes before TA is activated later during embryogenesis.
  27. Likewise, the statement that ALT might explain the high prevalence of cancer (lines 251/2) with age has to my knowledge absolutely no evidence, as ALT occurs VERY rarely, only in a very small subset of cancers and more frequently in sarcomas and in vitro.
  28. line 255/6: The statement about a potential increase of proliferation and cancer risk is not entirely true since it depends of the way TA is increased: in vitro by hTERT overexpression (which is only done in cultured cells) this is a constitutive, high overexpression. Other TA/hTERT modulations using either adenoviruses or natural telomerase activators do not bear such risks and only induce physiological amounts of telomerase extending the very shortest telomeres. Please amend this too oversimplified sentence.
  29. 264 "A different scenario happeneS..."
  30. 274: "p53 bound to THE subtelomere"
  31. line 307 the sentence "...which also involves in DDR" is wrong in grammar and ambiguous: what exactly referres "which" to- TRF, AKT or TGFbeta? and it should be "which (?) IS also INVOLVED in the DDR...
  32. line 315 (figure legend) should be: "EMT IS INVOLVED"
  33. figure 2: in the text just 1 shelterin: TRF2 just a bit of TRF1 is described while here now just "shelterin" is depictes as interaction partner. Its better to be more specific.

Reviewer 2 Report

In this review, Siti A. M. Imran et al. aim to discuss the known functions of EMT and telomere in ageing and the possible interaction between the two in age-related diseases.

This is an interesting approach, but it is not clear how it brings added value to the telomere field. There is no proper study directly discussed in the manuscript. It more an opinion paper, and it is up to the editor to assess whether he or she is interested in this type of publication.

Reviewer 3 Report

This review discusses EMT and telomers in aging and the possible interconnection between EMT and telomers in regulation of aging. This is a very interesting and appropriate topic. The paper is written well. There are however some minor issues that need to be corrected before publication:

-Abstract needs to state clearer what this review involves perhaps by using phrase such as “in this review…”

-There are some issues with abbreviations and full names. In many cases acronyms have not been defined in the first instance they used, for example: Line 34 (EMT), Line 38 (TGFB), Line 65 (ECM), Line 138 (a-SMA), Line 285 (ATM), Line 151 (IPF already defined in Line 149).

-Line 161, needs reference at the end of sentence

-Line 201 to 206 (“The Materials and Methods should be ….”) should be deleted. It appears it is part of the template.

-Some typo errors, eg. Line 265 (SNAIL1 controls), Line 266 (found to present), Line 267 (found to downregulate), Line 269 (should be TGF-β not TGF), Line 270 (plays), Line 302 (EMT4??), Line 305 (there instead of they), Line 308 (to cause), Line 309 (increase in the expression)

-Line 287, sentence doesn’t make sense. perhaps sentence shouldn’t end before “whereas”.

Round 2

Reviewer 1 Report

The authors addressed most of my major concerns such as modifying the title, but there is still quite some revision required, mainly regarding language, but also serious content-related issues, in particular in the area of telomeres, senescence and ageing where the authors are obviously not very familiar with. Considering there is absolutely no evidence about a connection between telomeres and EMT, it would make more sense to focus on associations which seem to exist-for example between senescence/inflammation and EMT. However, this side of senescence (in the form of SASP) is not necessarily connected since SASP also occurs during senescence of postmitotic cells where no telomere shortening and cell cycle arrest occur.

  1. abstract line 25: it should be "telomere protection"
  2. line 140: what does refer "former" to? It does not seem to be correct grammar
  3. line 188: remove comma after "repair"
  4. lines 189-191: sentence is redundant in content to above sentences and I do not understand what means "aging is a more complex and synergistic process"? More or synergistic compared to what? I also don't think that "synergistic" is a correct term here. Please best remove this sentence as no new information is given here.
  5. line 204: it should be "alpha"
  6. Please explain all abbreviations such as MS (line 245)
  7. lines 247/8 sentence is not correct in grammar since not the diseases are enriched
  8. line 255 "creates"
  9. lines 258, 262 and 265: remove "the" in front of "senescent" and "key"
  10. lines 260 and others: "expression" exclusively refers to GENES, so if you refer to proteins, please use "level/amount", but NOT expression
  11. IL-6 and 8 are pro-inflammatory SASP-factors connected to senescence. Consequently, it seems that SASP and inflammation promote EMT. If this is the case, its worth emphasising this interesting and important fact.
  12. line 270: remove "the" in front of "presenescent"
  13. same line: "fibroblasts of prostate fibroblast cells" does not make sense! Probably best to simply say "presenescent prostate fibroblasts".
  14. line 275 "increased level of insulin"
  15. line 277: "expression" see above
  16. line 301:"increased"
  17. line 303: statement requires reference
  18. line 310: "organisms"
  19. line 311: "tumors"
  20. The whole structure of the chapter "telomeres and aging" in confused, not logical and contains numerous wrong statements (see below for details).
  21. lines 318/19/20: remove "s" from telomere and write "shelterin" with a small letter
  22. 322: remove "the" in front of "shelterin"
  23. lines 324/5: please spell out the names of ALL shelterin components and correct RAP to repressor/activator protein.
  24. line 328/9 remove "the", in 329 also remove "will"
  25. line 330: replace "as" with "than" and add "and" between epithelial cells" and "are"
  26. line 333: replace "due to the telomere length in particular cells" with "due to critically short telomeres in somatic cells without telomerase activity".
  27. lines 336/7: shelterin does NOT maintain telomere length, but protects telomeres, only telomerase maintains/elongates telomeres in cells where it is active. The sentence in 337-9 repeats the same fact again and is, as the previous sentence not correct. There is a difference between maintaining/protecting telomeres and maintenance of telomere LENGHT. Only telomerase is able to do the latter.
  28. lines 339/340: Senescence occurs when telomeres are critically short, not just during any shortening. In addition, another outcome of telomere shortening in addition to senescence can be apoptosis.
  29. line 340: It is incorrect to state that senescence equals aging-both are related to different levels: senescence occurs at the cellular level while ageing is related to whole organisms. Please amend and correct this wrong statement!
  30. line 342: "radiation" has to be used in singular, similar to "damage"
  31. line 346/7/9: Please replace "activation" with "activity"
  32. line 347: Telomerase activity does NOT maintain telomere length via replication but just the opposite: during replication telomeres shorten due to the End replication problem, while telomerase adds telomere repeats DE NOVO, independent of replication! Please correct this wrong statement!
  33. what do you mean with "telomerase ACTIVITY remains low to none in the rest of the cells"? This is completely incorrect! What you probably want to say is "This process of telomere maintanance occurs only in cells with active telomerase while the majority of human somatic cells does not have telomerase activity". Also, reference 72 is not a good reference for this basic statement. Please find a better one.
  34. lines 349-354: it is completely unclear what you mean with "these cells" which in the sentence before you only described as the "rest of the cells" which is not a scientific category. Also: adult stem cells have not low, but inducible telomerase activity (Hiyama&Hiyama, 2007). Please correct this wrong statement.
  35. 349-51; As already explained above, not the slow and continuous process of telomere shortening prevents cell division and replication, but ONLY when this shortening reaches a minimal shortness which then activated a DNA damage response (DDR) which can result in either senescence or apoptosis.
  36. lines 352-355: Telomerase expression in human cells is not dynamic: it is constitutively expressed in germ cells and embryonic stem cells, can be up-regulated in adult stem cells, is not expressed in most human somatic cells, but present in lymphocytes and endothelial cells. Also, again: telomere shortening is NOT the same as senescence but the latter occurs ONLY when a critical short telomere length is reached. You completely oversimplify and falsify the complex relationship without providing any meaningful and true facts.
  37. lines 356/7: what do you mean with "confusing signals" or "purpose" of cells? all these are rather unscientific terms and do not describe scientific processes properly.
  38. the paragraph in lines 356-365 requires proper references!
  39. lines 360/1: Again, a highly unscientific description about "confusing cells" . Why would senescent cells "detect "other cells"?
  40. It is also complete nonsense that cells with telomerase would replace damaged or senescent cells-this is completely impossible! What you probably want to talk about is tissue regeneration. Senescence is irreversible and senescent cells cannot be replaced, thus it is important to prevent it before it happens. Senescent cells are persisting and cannot be replaced-this is exactly why ageing occurs!
  41. Lines 364/5: Also nonsense-telomeres are NOT able to regenerate CELLS or REPAIR damage! This is NOT what telomeres DO!!!!
  42. line 367: PBMCs are NOT cancer, please correct!
  43. lines 369+ The reason why cancer cells proliferate despite having short telomeres is that the have high levels of telomerase activity. Please add this important fact while ALT has barely any role for the majority of cancer and only is activated in very few specific types of it such as sarcomas.
  44. lines 386/7: Telomerase activation without other molecular events such as oncogene activation and inactivation of tumour suppressors by mutations does NOT induce cancer since stabilisation of telomeres on a background of a normal, stable karyotype just extends lifespan of cells (see Bodnar et al., 1998, Hahn WC et al., 1999).
  45. The whole structure of the chapter "telomeres and aging" in confused, not logical and contains numerous wrong statements (see below).
  46. line 397: please describe what you mean with "normal cells"
  47. line 399: please replace "was" with "were" since "cells" are plural!
  48. line 401: sentence requires a reference. Also, if you talk about TERRA you have to explain both the abbreviation and the content. What does Terra 2q, 11 and 18q mean?
  49. line 411: it should be "with telomeres" but, as already stated in the first review and as you corrected in the figure legend, it is mainly TRFs you provide connection to EMT with, so please amend this sentence accordingly as well.
  50. line 412: it should be "network prediction"
  51. line 418: remove "s" from "expression"-it is always used in singular only.
  52. line 423: "telomere protection" and "telomere end" would be correct, 
  53. line 439: A proper reference is required for the statement that "manipulating TRF2 levels induces tumorigenesis" since this might not be really true.
  54. For me the supposed relation between TRF2 and EMT/TGFbeta seems highly constructed with no single evidence provided and only pure speculations with no scientific basis.
  55. line 447: the statement again requires an appropriate reference.
  56. line 452: Do references 96 and 97 really show that telomere dysfunction increases EMT markers? In my view this is because TGFbeta can be a part of SASP and thus mediated by senescence, no matter how this process was initially caused, thus telomeres are involved completely indirectly. In my view, it would be better to focus the review on senescence, SASP and inflammatory factors and EMT than telomeres.
  57. line 454: remove duplicated "the"
  58. line 458: please make beta in TGFbeta in symbol font

Reviewer 2 Report

The amendments made by the authors are appropriate.

Two very recent and important papers, that are worth to be cited, could be added to the paragraph discussing the alt (page 6 line 254):

Hartlieb, S.A., Sieverling, L., Nadler-Holly, M. et al. Alternative lengthening of telomeres in childhood neuroblastoma from genome to proteome. Nat Commun 12, 1269 (2021). https://doi.org/10.1038/s41467-021-21247-8

and

de Nonneville, A., Reddel, R.R. Alternative lengthening of telomeres is not synonymous with mutations in ATRX/DAXX. Nat Commun 12, 1552 (2021). https://doi.org/10.1038/s41467-021-21794-0

Round 3

Reviewer 1 Report

The authors now addressed the majority of my comments satisfactorily and just very few changes in content and language are required.

  1. Please remove the article "the" in lines 175, 179, 181,184,186,213,214,218,229,230,266
  2. The shelterin protein TTP1 is NO tripeptidyltransferase one but a gene called ACD-I know its confusing, but that's what it is.
  3. In line 269: I think it should rather be "genome-wide sequencing" and please adapt the abbreviation accordingly.
